# High Throughput 3D Cell Migration Assay Using Micropillar/Microwell Chips

**DOI:** 10.3390/molecules27165306

**Published:** 2022-08-19

**Authors:** Sang-Yun Lee, Lily M. Park, Yoo Jung Oh, Dong Hyuk Choi, Dong Woo Lee

**Affiliations:** 1Department of Biomedical Engineering, Gachon University, Seongnam 13120, Korea; 2Cytek Biosciences, 47215 Lakeview Blvd, Fremont, CA 95348, USA; 3Department of Biomedical Engineering, Konyang University, Daejeon 35365, Korea; 4Medical & Bio Decision (MBD), Suwon 16229, Korea

**Keywords:** migration assay, invasion assay, 3D cell culture, high throughput screening (HTS), high contents screening (HCS)

## Abstract

The 3D cell migration assay was developed for the evaluation of drugs that inhibit cell migration using high throughput methods. Wound-healing assays have commonly been used for cell migration assays. However, these assays have limitations in mimicking the in vivo microenvironment of the tumor and measuring cell viability for evaluation of cell migration inhibition without cell toxicity. As an attempt to manage these limitations, cells were encapsulated with Matrigel on the surface of the pillar, and an analysis of the morphology of cells attached to the pillar through Matrigel was performed for the measurement of cell migration. The micropillar/microwell chips contained 532 pillars and wells, which measure the migration and viability of cells by analyzing the roundness and size of the cells, respectively. Cells seeded in Matrigel have a spherical form. Over time, cells migrate through the Matrigel and attach to the surface of the pillar. Cells that have migrated and adhered have a diffused shape that is different from the initial spherical shape. Based on our analysis of the roundness of the cells, we were able to distinguish between the diffuse and spherical shapes. Cells in Matrigel on the pillar that were treated with migration-inhibiting drugs did not move to the surface of the pillar and remained in spherical forms. During the conduct of experiments, 70 drugs were tested in single chips and migration-inhibiting drugs without cell toxicity were identified. Conventional migration assays were performed using transwell for verification of the four main migration-inhibiting drugs found on the chip.

## 1. Introduction

Migration is a critical property of live cells for normal cell development and immune responses as well as other pathological processes, including disease progression, cancer metastasis, and inflammation [1,2,3]. Migration that has progressed to an invasive condition contributes rather significantly to the survival rate of cancer patients. Thus, methods used for the examination of cell migration and invasion are very useful and important in a wide range of cancer studies. Many in vitro conventional assays are used in the study of cell migration; these include transwell [4] and wound healing [5] methods. Although these methods are widely known and have advantageous features, mainly easy accessibility, they lack the capacity to mimic the in vivo characteristics of actual human cancer. Keeping in mind that these assays are based on two-dimensional cell monolayer culture where some cells lose their phenotypic properties, there is a need for the development of a cancer migration assay that enables the growth of cells in three-dimensional formations. Measurement of cell viability to check for inhibition of cell migration without cell toxicity was not possible using wound-healing and transwell methods.

Several 3D invasion assays [6,7,8] were introduced in an effort to overcome these issues. Cells were cultured in an extracellular matrix and the spread of cells from the tumor sphere was observed. Successful cell migration was observed under three-dimensional cell culture conditions; analytical throughput was low due to the complex 3D cell culture process and difficulty in obtaining and analyzing images of cells that spread from the tumor area. A micropillar and microwell chip platform for 3D cell-based high throughput screening was previously developed by our group [9,10,11]. Measurement of the various 3D cell responses of many drugs is based on the 3D cell area exposed to drugs. 

In this paper, we propose a 3D cell migration assay using a micropillar/microwell chip in order to simplify the 3D cell culture process and to measure cell migration by the analysis of cell roundness in a Matrigel spot (Figure 1a). Initially, we collected the cells at the curved end of the Matrigel spot. During culture, pillar and well chips were inverted. Over time, cells migrate through the Matrigel and attach to the surface of the pillar by gravity, as shown in Figure 1a. Cells that have migrated and adhered have a diffused shape that is different from the initial spherical shape. Based on our analysis of the roundness of the cells, we were able to distinguish between the diffuse and spherical shapes. Cells in Matrigel on the pillar that were treated with migration-inhibiting drugs did not move to the surface of the pillar and remained in spherical forms, as shown in Figure 2. Thus, cell migration and viability can be measured by simultaneously analyzing the roundness and area of cells using the 3D cell migration assay. During the conduct of experiments, 70 drugs were tested in single chips and migration-inhibiting drugs were identified. To prove the proposed 3D cell migration assay, conventional migration assays were performed using transwell for verification of the four migration-inhibiting drugs which were found in the proposed 3D cell migration assay.

## 2. Materials and Methods

### 2.1. Cell Preparation

The A549 lung cancer cell line was obtained from ATCC and cultured in the Rosewell Park Memorial Institute (RPMI) medium 1640 (CellGro, New York, NY, USA), supplemented with 10% fetal bovine serum (FBS; CellGro, Mexico City, Mexico) and 1% penicillin–streptomycin (Invitrogen, Grand Island, NE, USA) in cell culture flasks (T-75 and T-175; Corning, CA, USA) in a humidified 5% CO_2_ incubator at 37 °C. The cells were routinely passaged every three days at 70% confluence. For the preparation of the cell suspensions, a confluent layer of cells was trypsinized with 0.3 mL of 0.25% trypsin in 0.53 mM EDTA (Invitrogen) from the culture dish, and the cells were then resuspended in 7 mL of RPMI supplemented with 10% FBS and 1% penicillin–streptomycin. Following centrifugation at 2000 rpm (700 *g*) for 3 min, the supernatant was removed, and the cells were resuspended with RPMI supplemented with 10% FBS to a final concentration of 10 × 10^6^ cells/mL. The number of cells in RPMI was calculated using the AccuChip (Digital Bio, Seoul, Korea) automatic cell counting kit. For the passage of the cells, 1 × 10^6^ cells were seeded in a T-75 flask containing 15 mL of growth medium.

The A549 lung cancer cell line was obtained from ATCC and cultured in the Rosewell Park Memorial Institute (RPMI) medium 1640 (CellGro, New York, NY, USA), supplemented with 10% fetal bovine serum (FBS; CellGro, Mexico City) and 1% penicillin–streptomycin (Invitrogen, Grand Island, NE, USA) in cell culture flasks (T-75 and T-175; Corning, CA, USA) in a humidified 5% CO_2_ incubator at 37 °C. The cells were routinely passaged every three days at 70% confluence. For the preparation of the cell suspensions, a confluent layer of cells was trypsinized with 0.3 mL of 0.25% trypsin in 0.53 mM EDTA (Invitrogen) from the culture dish, and the cells were then resuspended in 7 mL of RPMI supplemented with 10% FBS and 1% penicillin–streptomycin. Following centrifugation at 2000 rpm (700 *g*) for 3 min, the supernatant was removed, and the cells were resuspended with RPMI supplemented with 10% FBS to a final concentration of 10 × 10^6^ cells/mL. The number of cells in RPMI was calculated using the AccuChip (Digital Bio, Seoul, Korea) automatic cell counting kit. For the passage of the cells, 1 × 10^6^ cells were seeded in a T-75 flask containing 15 mL of growth medium.

### 2.2. Preparation of Micropillar and Microwell Chips

The micropillar and microwell chips are made by plastic injection molding; this system is more robust and flexible for use in compound screening [9,10,11]. Polystyrene, a widely used biocompatible plastic, is used in the preparation of the micropillar and microwell chips. The micropillar chip contains 532 micropillars (0.75 mm pillar diameter and 1.5 mm pillar-to-pillar distance). For polymerization of the Matrigel, the surface of the pillar was 10 s plasma-treated (80 W power, 5 × 10^−4^ Torr using air) and stamped with diluted laminin solution (L2020-1 mg, Sigma, St. Louis, MO, USA) in PBS. For the preparation of the mixture of Laminin coating solution, 10 mL of PBS was mixed with a 1/100 ratio of pure laminin solution (1 mg/mL). The microwell chip also contained 532 microwells (1.2 mm well diameter and 1.5 mm well-to-well distance). Both the micropillar chip and the microwell chip were similar to conventional microscopic glass slides in terms of size (75 × 25 mm). The SODIC PLUSTECH injection molder was used in the performance of plastic molding; the fabricated chips are shown in Figure 1b.

### 2.3. Chip Layout and Experimental Procedure

The chip layout for screening the 70 compounds in a single micropillar or microwell chip is shown in Figure 3a. Six replicates in the case of migration and migration inhibition are shown in Figure 2b. 

In the micropillar chip, approximately 80 cells were immobilized with Matrigel ECM solution in each micropillar. The microwell chip was divided into 72 regions; each region contained six replicates, as shown in Figure 3a. Each region contained one drug with the same dosage (20 μM). The migration assay is a short-term assay which cultures cells for 24 h. We, therefore, chose a 20uM high concentration in all compounds, and considered that cells embedded in Matrigel are generally highly resistant to compounds. Growth media with the same volume of DMSO as the other wells with drugs was used as the control for all 70 drugs, which were diluted in DMSO (final concentration is 0.05% *v*/*v*). A 50 nL mixture of cell/Matrigel was dispensed on the micropillar using the Cell Spotter (ASFA™ spotter, Medical & Bio Decision, Suwon, Korea). Using a solenoid valve (Lee Company, Westbrook, CT, USA), the Cell Spotter dispensed 50 nL droplets of the cell/Matrigel mixture on the micropillar and 950 nL of media or drugs in the microwell, respectively. The micropillar chip was combined with an empty microwell chip and the combined chips were placed in a refrigerator (4 °C) to allow the cells to move to the tip of the Matrigel spot. Using this chilling process, the cells move through the Matrigel as it has not yet gelled. After 15 min, the combined chips move to the incubation to gelling Matrigel; gelation time is 15 min. The micropillar chip is then separated from the empty microwell chip and moved to a new microwell chip filled with media and drugs. Following incubator of cells with drugs for 24 h, the live cells in the Matrigel spot are stained with 0.5 μM Calcein AM (C3099, Invitrogen, CA, USA). Images of cells were obtained using an optical fluorescence scanner (ASFA™ scanner, Medical & Bio Decision, Suwon, Korea). Green dots represent stained live cells, as shown in Figure 2, Figure 3 and Figure 4. 

### 2.4. Cell Staining

Following incubation, the cells were stained with Calcein AM (C3099, Invitrogen, USA) (Figure 1c), which can be transported through the cellular membrane and can produce a green fluorescence response from living cells, making it useful for visualizing cell morphology of live cells. All 532 micropillars on the micropillar chip separated from the microwell chip were immersed in a new microwell chip filled with PBS buffer and staining dye solution. The micropillar chip was washed twice for 5 min each by immersing micropillars with cell spots in a new microwell chip filled with PBS buffer. For the preparation of the staining dye solution, 1.0 µL of Calcein AM (4 mM stock from Invitrogen) was added to 8 mL of PBS buffer. For staining the cell spots, 950 nL of the dye solution was dispensed in the microwell chip and the micropillar chip was then combined with the microwell chip. After incubating the micropillar/microwell chip in darkness for 45 min at room temperature, the micropillar chip was washed twice for 15 min each by combining it with a microwell chip filled with PBS buffer to remove the excess dye in the Matrigel spots. After drying the micropillar chip in darkness for at least 2 h, imaging of the entire micropillar chip was performed using an optical fluorescence scanner (ASFA™ scanner, Medical & Bio Decision, Suwon, Korea) equipped with a mercury light source (Olympus U-RFL-T), a ccd camera (Point Gray, Nashville, TN, USA), 4× and 10× objective lenses (Olympus UPlanFLM, TN, USA), and excitation/emission filters (Thorlab, Nashville, TN, USA) to detect the location of each cell spot exposed to a drug [10]. An excitation filter of 475 ± 35 nm and an emission filter of 530 ± 43 nm, which are optimum for detecting green fluorescence from live cells stained with Calcein AM, were used. The microscope located on a moving stage in the scanner focused automatically on cell spots by moving in the z direction and selecting the highest fluorescent cell image; 532 individual pictures were taken from a single stained micropillar chip at 4× magnification. The 532 pictures of cell spots were then consolidated into a single JPEG image for data analysis (Figure 3a).

### 2.5. Cell Image Analysis for the Migration Assay

For the 3D cell migration assay, the performance of the multiparameter analysis from cell images is important. Cells seeded in Matrigel have a spherical form. Over time, cells migrate through the Matrigel and attach to the surface of the pillar. Cells that have migrated and adhered have a diffused shape that is different from the initial spherical shape. Based on our analysis of the “roundness” of cells, we were able to distinguish between the diffuse and spherical shapes. Cells in Matrigel on the pillar that were treated with migration-inhibiting drugs did not move to the surface of the pillar and remained in spherical forms. In the case of highly toxic drugs, the damaged cells were not able to move and slowly disappeared. Both highly toxic drugs and drugs that inhibit cell migration cause cells to take a spherical form. However, the area of the cell is different. A small cell area and low viability are observed for damaged cells, as shown in Figure 2a. Therefore, the “roundness” and “area” of cells were extracted from the cell image and used to calculate cell migration and viability, respectively. The object area and convex hull area of each cell were measured for calculation of roundness, as shown in Figure 2a. The viability was calculated according to the area of the cell. Cell Analyzer software (Medical & Bio Decision, Seoul, Korea) was used for calculation of the roundness and viability of cells.

### 2.6. Statistical Analysis

One-way ordinary ANOVA was used to distinguish drugs of low roundness compared to the control. The control and drug have six replicate points and are statistically analyzed as mean and standard deviation. We tested three chips in the same condition. The *p*-values are calculated using three mean values from three chips. All statistical analyses were performed using GraphPad Prism software 9 (9.0 version, Los Angeles, CA, USA). Statistical analysis of roundness of drug was compared to control (no drug). The *p*-value was set at a confidence interval of 99%, two-tailed. A *p*-value less than 0.01 was considered statistically significant.

## 3. Results and Discussion

### 3.1. Area-Roundness Plot for Evaluation of Cell Migration

The diffuse and spherical shapes of cells are shown in Figure 2b. The roundness and area of cells were calculated from six spot images and graphed as shown in Figure 2c,d. The closer roundness is to zero, the closer it is to a circle. The roundness of the circle is 0. Thus, a high roundness value indicates a migrating spread of cells. Thus, high roundness means that the cells moved to the pillar, indicating cell migration. The individual cell area-roundness plots are shown in Figure 2c,d. Single cells are extracted from spot images. The area and roundness of a single cell were calculated. Figure 2c,d shows the individual cell area-roundness plot explained in Figure 2a. As shown in the individual cell area-roundness plot of Figure 2a, none of the effective drugs caused damage to cells. The cells moved to the surface of the pillar and formed a diffused shape. Thus, those cells showed high roundness and area. Therefore, the cells showing high roundness and area were located in quadrant 1 of the cell area-roundness plot. The high roundness mean cell has a diffused shape, so, there are no cells with high roundness and small cell area in quadrant 2. Highly toxic drugs cause damage to cells. The area and roundness of the damaged cells were reduced, and those cells were located in quadrant 3 of the cell area-roundness plot. The drugs that inhibited migration without cell toxicity mainly reduced the roundness of cells; those cells were located in quadrant 4 of the cell-roundness plot, as shown in the individual cell area-roundness plot of Figure 2a. Thus, the drugs that inhibited migration could be identified in the individual cell area-roundness plot. Under no drug condition, cells moved easily to the pillar and many cells showed high roundness and cell area, as shown in Figure 2c. Seventeen cells in quadrant 1 of Figure 2c had roundness greater than 0.2 and cell area greater than 500 pixels. These cells having high roundness and cell area indicate migrated cells. Under migration inhibition drug conditions, there were two migrated cells (roundness greater than 0.2 and cell area greater than 500 pixels) in quadrant 1 of Figure 2d. The number of cells in quadrant 1 of the cell-roundness plot was significantly reduced compared to the no drug condition (control). 

### 3.2. Cell Migration Inhibition Screening of 70 Target Drugs

Seventy target drugs were tested for validation of the high throughput 3D cell migration assay. The roundness and area of individual cells were calculated spot by spot on the micropillar chip, as shown in Figure 3a. One drug had six replicated spots. Cell area and roundness in 70 cancer drugs (including two controls) are shown in Table 1. The average and standard deviation of the roundness were calculated according to the average roundness of individual cells in six spots. The average roundness calculated at a spot extracts the top 10% of roundness for the individual cells at that spot. The average and standard deviations of the cell area were calculated according to six sum cell areas in six spots. The normalized cell area-roundness plot for 70 drugs is shown in Figure 3b. #1 and #2 are the control without drug. Most drugs were not located in an effective region (quadrant 1). Drugs #32, #52, and #11 showed high toxicity due to low cell area and low roundness. Four drugs (#3, #7, #38, and #39) showed strong inhibition of cell migration without high cell toxicity. Compared to the roundness of the control condition (no drugs), *p*-values of 70 drugs were calculated as shown in Figure 3c. A *p*-value lower than 0.01 indicates significantly reduced roundness and inhibition of cell migration. Among five drugs having the smallest P-value, inhibition of cell migration without cell toxicity was demonstrated scientifically for four drugs (#3, #7, #38, #39). The 70 drugs and their target are shown in Table 1. Inhibition of cell migration by AEE788 (#3), Dacomitinib (#7), Cabozantinib (#38), and Foretinib (#39) was previously demonstrated using the wound-healing and transwell assays [12,13,14,15]. The individual cell area-roundness plots of representative migration and migration inhibition conditions are shown in Figure 4. Compared to the control (no drug), no migration inhibition was observed for Erlotinib, PF-044449913, and Vandetanib. However, a reduction in the number of migrated cells was observed in quadrant 1 of the cell area-roundness plot for drugs (#3, #7, #38, #39). Cell images and transwell images of representative migration and migration inhibition conditions are shown in Figure 5. Four drugs (#3, #7, #38, and #39) that inhibited cell migration in the proposed 3D cell-based migration assay prevented penetration of cells into the transwell membrane.

## 4. Conclusions

Using a micropillar/microwell chip, the 3D cell migration assay was developed for the evaluation of drugs that inhibit cell migration using high throughput methods. For measurement of cell migration, cells were encapsulated with Matrigel on the surface of the pillar, and the morphology of cells attached to the pillar through Matrigel was analyzed. The micropillar/microwell chips contained 532 pillars and wells, which analyze cell roundness and size, respectively, for measurement of cell migration and viability. Cells seeded in Matrigel have a spherical form. Over time, cells migrate through the Matrigel and attach to the surface of the pillar. Cells that have migrated and adhered have a diffused shape that is different from the initial spherical shape. Based on our analysis of the roundness of the cells, we were able to distinguish between the diffuse and spherical shapes. Cells in Matrigel on the pillar that were treated with migration-inhibiting drugs did not move to the surface of the pillar and remained in spherical forms. During the conduct of experiments, 70 drugs were tested in single chips. AEE788 (#3), Dacomitinib (#7), Cabozantinib (#38), and Foretinib (#39), which inhibited cell migration without cell toxicity, were identified and conformed with the transwell assay. Thus, the proposed high throughput 3D cell migration assay could be used to evaluate the cell migration inhibition of drugs considering cell toxicity by analysis of 3D cell morphology.

## Figures and Tables

**Figure 1 molecules-27-05306-f001:**
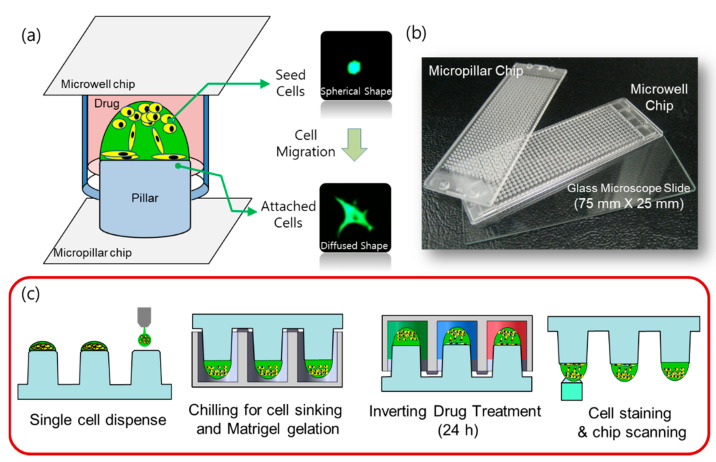
High throughput 3D cell migration assay. (**a**) A schematic view of cell migration in micropillar/microwell chips. (**b**) Photographs of the micropillar/microwell chip. (**c**) Experimental procedure for the 3D cell migration assay. 0.5 μL spots of cells and Matrigel were dispensed on the micropillar. The micropillar was combined with the microwell and placed on ice (4 °C) so that cells descended to the bottom of the spot. Chips were then placed in an incubator (37 °C) for 5 min for Matrigel gelation. The micropillar chip containing cells was combined with a new microwell chip filled with drugs and incubated for 24 h. After staining cells with Calcein AM, scanning of the cells on the micropillar was performed using an automatic fluorescence microscope.

**Figure 2 molecules-27-05306-f002:**
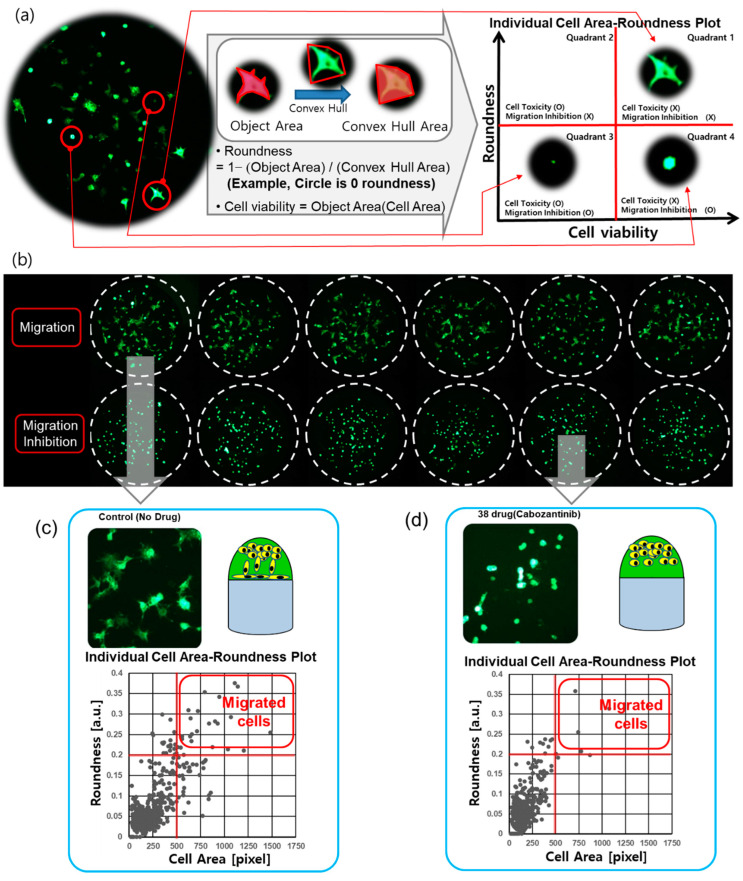
Cell morphology analysis for the cell migration assay. (**a**) Cell image of a spot on a micropillar (no drug condition). Different morphology of cells was analyzed according to the roundness. The migrating and attached cells had a diffused shape, while the initial seed cells had a spherical shape. The cells showing high roundness and area were located in quadrant 1 of the cell area roundness plot. The high roundness mean cell has a diffused shape, therefore, there are no cells with high roundness and small cell area in quadrant 2. The area and roundness of the damaged cells were reduced, and those cells were located in quadrant 3 of the cell area-roundness plot. The drugs that inhibited migration without cell toxicity mainly reduced the roundness of cells; those cells were located in quadrant 4 of the cell-roundness plot. (**b**) Six replicated spot images at migration cells and migration inhibition cells. (**c**) Individual cell area-roundness plot under no drug condition. (**d**) Individual cell area-roundness plot for the cell migration inhibition drug (Cabozantinib).

**Figure 3 molecules-27-05306-f003:**
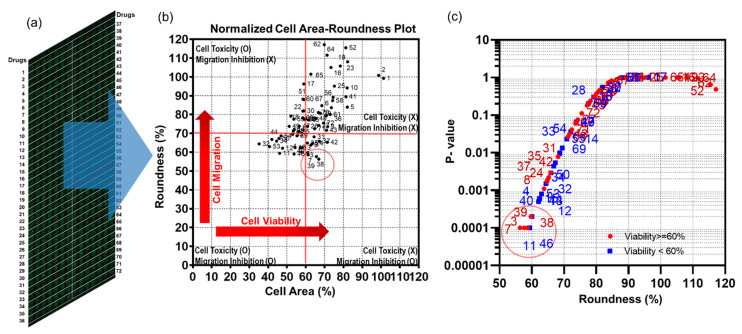
Migration assay for 70 drugs using a single micropillar/microwell chip. (**a**) full scanning image of 532 micropillar chips containing 70 drugs, including two controls (no drug). Each drug was tested in seven replicates. (**b**) The mean cell area-roundness graph shows four areas: no cell toxicity/no cell migration inhibition, no cell toxicity/cell migration inhibition, cell toxicity/cell migration inhibition, cell toxicity/no cell migration inhibition. (**c**) *p*-values of the roundness compared to control (no drug) according to average roundness.

**Figure 4 molecules-27-05306-f004:**
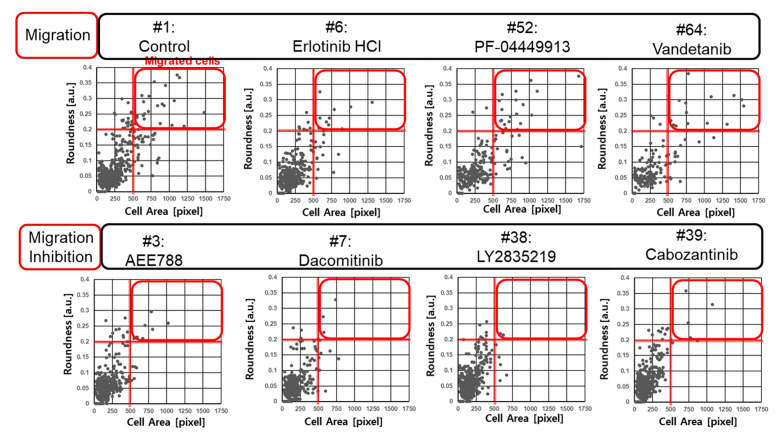
Individual cell area-roundness graphs of four migration conditions and four migration inhibition conditions.

**Figure 5 molecules-27-05306-f005:**
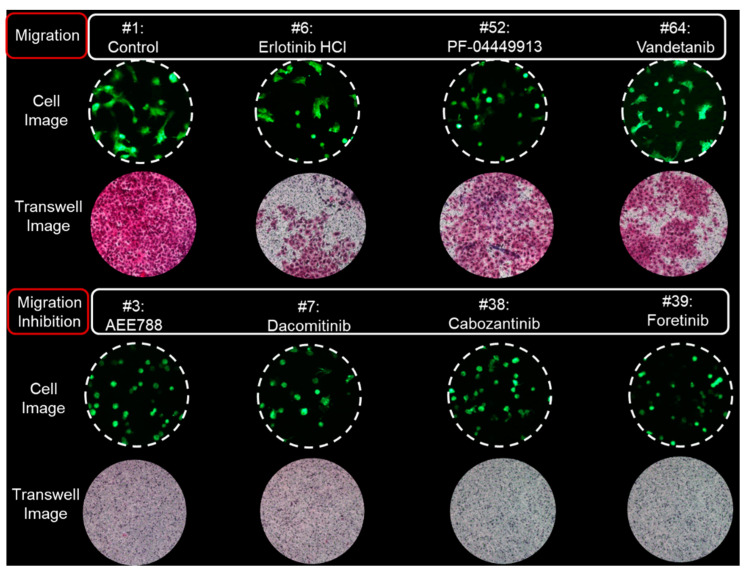
Cell images in spots on the micropillar chip and cell images in the membrane of the transwell.

**Table 1 molecules-27-05306-t001:** Summary of cell area and roundness in 70 cancer drugs (including two controls). * These drugs are migration inhibition drugs without cytotoxicity.

	Targeted Anticancer Drug	Target	Chip Number #1		Chip Number #2		Chip Number #3		Average		
			Cell Area [%]	Round-ness [%]	Cell Area [%]	Round-ness [%]	Cell Area [%]	Round-ness [%]	Cell Area [%]	Round-ness [%]	*p*-Value
1	DMSO	-	94.79	103.36	100.88	94.29	108.20	99.99	101.29	99.21	-
2	DMSO	-	105.21	96.64	99.12	105.71	91.80	100.01	98.71	100.79	-
3 *	AEE788 (NVP-AEE788)	EGFR	69.06	55.50	62.50	55.89	65.38	61.66	65.65	57.68	0.0001
4	Afatinib (BIBW2992)	EGFR	54.61	57.69	52.01	61.95	67.68	67.80	58.10	62.48	0.0006
5	BMS-599626 (AC480)	EGFR	81.63	73.69	76.44	78.58	88.11	99.62	82.06	83.96	0.75
6	Erlotinib HCl	HER1/EGFR	72.96	72.69	80.02	90.46	67.76	87.89	73.58	83.68	0.7216
7 *	Dacomitinib (PF299804, PF-00299804)	EGFR	57.76	55.84	56.03	55.60	69.99	64.49	61.26	58.64	0.0001
8 *	Gefitinib (Iressa)	EGFR	60.62	57.17	60.45	58.81	67.12	75.38	62.73	63.79	0.0011
9	Lapatinib	EGFR	67.08	57.42	72.09	85.04	75.61	78.18	71.59	73.55	0.0603
10	Neratinib (HKI-272)	EGFR	76.44	94.96	79.24	93.30	90.27	94.22	81.98	94.16	0.9992
11	CI-1033 (Canertinib)	EGFR, HER2	38.26	54.89	47.97	55.21	52.14	68.18	46.13	59.43	0.0001
12	CO-1686	EGFR	40.62	58.73	53.64	62.96	48.69	64.64	47.65	62.11	0.0005
13	BKM120 (NVP-BKM120)	PI3K	54.91	58.80	55.38	58.03	60.85	70.24	57.05	62.36	0.0006
14	BYL719	PI3K	39.10	66.13	58.45	75.64	64.15	74.42	53.90	72.06	0.036
15	XL147	PI3K	48.48	55.73	54.08	59.90	60.23	71.83	54.26	62.49	0.0006
16	Everolimus (RAD001)	mTOR	60.65	95.53	89.76	137.26	69.68	82.25	73.36	105.01	0.999
17	AZD2014	mTOR	48.24	96.33	63.74	100.02	64.85	92.44	58.94	96.27	0.9996
18	PF-05212384 (PKI-587)	PI3K/mTOR	41.57	67.66	60.59	90.06	54.02	79.78	52.06	79.17	0.3033
19	XL765 (SAR245409)	PI3K/mTOR	80.57	105.60	82.54	106.67	71.79	105.11	78.30	105.79	0.9988
20	BEZ235	PI3K/mTOR	79.75	104.83	88.85	132.12	63.90	125.80	77.50	120.92	0.1993
21	AZD5363	Akt1/2/3	125.69	188.76	127.99	184.57	72.51	149.46	108.73	174.26	<0.0001
22	ABT-199 (GDC-0199)	Bcl-2	56.94	81.25	67.70	79.67	50.74	84.60	58.46	81.84	0.5344
23	ABT-888 (Veliparib)	PARP	82.36	99.89	93.57	120.18	70.65	104.00	82.20	108.02	0.9983
24 *	AUY922 (NVP-AUY922)	HSP (e.g., HSP90)	65.79	57.14	67.87	62.98	57.08	74.03	63.58	64.72	0.0017
25	Axitinib	VEGFR1/2/3, PDGFRβ and c-Kit	87.19	85.53	77.38	107.06	61.00	92.71	75.19	95.10	0.9994
26	AZD4547	FGFR1/2/3	51.78	64.85	62.87	72.44	68.26	83.80	60.97	73.70	0.0633
27	AZD6244 (Selumetinib)	MEK1	65.14	72.99	62.75	69.57	72.98	80.13	66.96	74.23	0.0755
28	LGK-974	PORCN	46.90	74.60	65.73	74.46	65.42	83.49	59.35	77.52	0.1995
29	BGJ398 (NVP-BGJ398)	FGFR1/2/3	64.52	78.02	76.15	82.44	62.56	81.34	67.74	80.60	0.418
30	Bortezomib (Velcade)	proteasome	50.72	72.65	64.43	86.97	61.71	86.31	58.95	81.98	0.5476
31 *	Cediranib (AZD2171)	VEGFR, Flt	68.77	68.14	63.70	62.38	60.78	73.95	64.42	68.16	0.0078
32	Crizotinib (PF-02341066)	Met, ALK	27.10	55.61	34.59	61.27	43.96	76.35	35.22	64.41	0.0015
33	Dasatinib (BMS-354825)	Bcr-Abl	45.84	60.08	52.31	67.96	60.87	80.35	53.01	69.46	0.0134
34	Dovitinib (TKI-258)	Flt3, c-Kit, FGFR1/3, VEGFR1/2/3, PDGFRα/β	39.31	64.12	44.12	61.21	50.03	72.33	44.49	65.89	0.0029
35 *	Imatinib (Gleevec)	v-Abl, c-Kit, and PDGFR	60.99	56.33	58.13	64.82	63.48	73.51	60.87	64.89	0.0019
36	INCB28060	Met	75.04	85.68	54.00	62.89	83.63	89.35	70.89	79.31	0.3133
37 *	LY2835219	CDK4/6	65.04	63.88	61.82	61.12	74.88	72.56	67.25	65.85	0.0029
38 *	Cabozantinib (XL184)	VEGFR2,c-Met, Ret, Kit, Flt-1/3/4, Tie2, and AXL	72.29	53.61	61.67	53.81	66.80	61.60	66.92	56.34	0.0001
39 *	Foretinib (XL880)	HGFR and VEGFR, mostly for Met and KDR	68.42	62.34	54.61	51.91	60.14	65.09	61.06	59.78	0.0002
40	Ibrutinib	Btk, modestly potent to Bmx, CSK, FGR, BRK, HCK	64.18	57.45	56.69	61.32	52.83	61.29	57.90	60.02	0.0002
41	Vemurafenib	B-Raf V600E	89.30	87.57	73.61	85.28	81.12	95.50	81.34	89.45	0.998
42 *	Trametinib	MEK1/2	72.87	63.45	65.34	58.27	76.21	74.03	71.48	65.25	0.0022
43	LDE225 (NVP-LDE225,Erismodegib)	smoothened	71.75	67.78	66.42	67.76	74.75	79.47	70.97	71.67	0.0312
44	LDK378	ALK	40.91	71.54	40.08	52.30	45.02	76.56	42.00	66.80	0.0044
45	LEE011	CDK4/6	64.46	68.06	58.82	76.17	61.15	87.17	61.48	77.13	0.18
46	Nilotinib (AMN-107)	Bcr-Abl	56.55	57.67	55.32	56.94	49.39	62.52	53.75	59.04	0.0001
47	Olaparib (AZD2281)	PARP1/2	52.88	65.03	59.36	79.15	57.27	77.26	56.50	73.81	0.0658
48	Panobinostat (LBH589)	HDAC	65.80	72.57	68.79	85.28	61.72	75.96	65.44	77.94	0.223
49	Pazopanib HCl	VEGFR1/2/3, PDGFR, FGFR, c-Kit	49.69	60.19	58.62	81.79	45.25	79.23	51.18	73.74	0.0642
50	PD 0332991 (Palbociclib HCl)	CDK4/6	41.81	62.29	52.55	66.70	42.79	72.85	45.72	67.28	0.0054
51	PF-04449913	HSP90	56.63	71.01	67.66	109.08	52.38	83.74	58.89	87.94	0.9818
52	Sotrastaurin (AEB071)	PKC	72.33	110.89	96.21	118.09	75.36	117.47	81.30	115.48	0.6462
53	Sunitinib Malate (Sutent)	VEGFR2 and PDGFRβ	30.35	59.15	48.48	61.27	43.38	68.56	40.74	62.99	0.0008
54	Tandutinib (MLN518)	FLT3, PDGFR, and KIT	38.86	66.33	52.07	65.61	49.69	73.96	46.88	68.63	0.0096
55	Tivozanib (AV-951)	VEGFR, c-Kit, PDGFR	56.42	80.38	53.08	56.41	59.65	75.81	56.38	70.87	0.0232
56	Vismodegib (GDC-0449)	Smoothened homologue (SMO)	62.85	86.17	89.86	89.25	70.88	92.06	74.53	89.16	0.992
57	PHA-665752	c-Met inhibitor	N/A	N/A	N/A	N/A	N/A	N/A	N/A	N/A	N/A
58	Dabrafenib	BRAFV600	72.19	79.85	88.35	87.54	62.29	94.79	74.27	87.39	0.9783
59	Regorafenib	VEGFR1/2/3, PDGFRβ, Kit, RET and Raf-1	35.18	68.49	66.17	79.01	59.54	85.13	53.63	77.54	0.201
60	Bosutinib	dual Src/Abl	52.73	84.80	62.73	86.68	60.19	93.09	58.55	88.19	0.9828
61	Carfilzomib	proteasome	77.84	76.12	69.89	84.25	71.24	79.79	72.99	80.05	0.3711
62	Ruxolitinib	JAK1/2	55.32	119.28	78.83	123.65	75.30	108.41	69.82	117.11	0.4832
63	Vandetanib	VEGFR2	44.57	84.42	66.13	78.01	64.67	72.19	58.45	78.21	0.2388
64	TMZ	alkylating agent	48.11	113.45	92.32	118.08	73.51	103.04	71.31	111.52	0.9624
65	Amorolfine	morpholine antifungal drug	45.15	111.07	78.28	95.52	64.67	97.78	62.70	101.46	0.9997
66	Mevastatin	HMG-CoA reductase inhibitor	59.74	68.48	60.21	78.15	60.06	86.09	60.00	77.57	0.2027
67	Amiodarone	antiarrhythmic medication	67.57	98.77	73.39	69.44	64.55	85.92	68.51	84.71	0.8207
68	Fluvastatin Na	anticholesterol agent; HMGCoA inhibitor	36.69	63.96	66.91	69.80	56.38	79.94	53.33	71.23	0.0265
69	Mycophenolic acid	inosine-5′-monophosphate dehydrogenase inhibitor	50.11	65.49	57.27	63.37	58.64	77.28	55.34	68.71	0.0099
70	Raloxifene HCl	estrogen receptor inhibitor	65.00	72.15	62.60	81.25	72.44	90.00	66.68	81.13	0.4663
71	Astemizole	histamine receptor ligand	56.49	69.10	59.70	67.00	77.90	81.21	64.69	72.44	0.0412
72	Fenretinide	retinoic acid receptor ligand	70.80	74.34	49.86	62.30	87.74	89.79	69.47	75.48	0.1116

## Data Availability

Not applicable.

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
