# Peer review of "High Throughput 3D Cell Migration Assay Using Micropillar/Microwell Chips"

_molecules, 2022, doi:10.3390/molecules27165306_

Round 1

Reviewer 1 Report

This manuscript describes a method for high throughput evaluation of drugs. The method allows evaluation of the effect on cell viability and migration of drug treatments. The authors have developed a micropillar/microwell system that, in combination with calcein staining and image analysis, can evaluate the effect of drugs on cell morphology and hence migration and viability. They screened 70 drugs and identified migration inhibiting drugs. Four of these drugs were validated using conventional migration assays.

Questions of interest:

Why do the cells migrate through the matrigel towards the pillar? Does it have anything to do with gravity?

Why do the chips need to be inverted for drug treatment?

Line 114 seems to imply that all the drugs were added at 20uM, why would this be the case?

The manuscript is mostly well written and clearly describes the work carried out. The data presentation and figure preparation are excellent. The figures really aid understanding. In order to make the meaning even easier to understand some changes are recommended:

The introduction would benefit from the use of some paragraphs

Lines 44-50 - Due to use of tenses and phrasing used it's difficult to understand if this is previous work in the field, generally known facts or opinion

Line 56-57 - The meaning of this sentence is unclear

Line 68 - Does this mean four migration-inhibiting drugs that were already known? Four classes of migration-inhibiting drugs? Four of the migration-inhibiting drugs with the most easy to observe effects?

Figure 1 - Replace "Slide Glass" with "Glass microscope slide". I suggest changing "Icing for cell down And Matrigel Gelation" with "Chilling for cell sinking and Matrigel gelation". In the legend it might be helpful to explain what the cells were stained with

Methods

Line 88 - centrifuge speed is reported in rpm, this should be reported in g

Line 99 - Please give details of plasma treatment

Line 101 - Please report final concentration of laminin. Where was the laminin from?

Line 115 - please give details of final DMSO concentration in growth media

Line 121 - Icing means "a sweet, creamy spread, as of confectioners' sugar, butter, and flavouring, for covering cakes, cookies" I suspect "chilling" would be a better word to use. "migrate" also does not seem to be a good word to use as it tends to suppose an active process, whereas the cells probably fall through gel to settle at the bottom

Line 123 - replace incubation with incubator

Line 126 - please give supplier and concentration of calcein AM used

Line 157-158 - Why was the chip dried for 2 hours after staining? This seems like a long time for a live cell stain to remain on cells

Results and Discussion

Line 199 - Would suggest using "high roundness value" as "high roundness" is confusing. To me it means that highly rounded cells have migrated, not that cells with a high value for roundness have migrated

Line 202 - Replace "spread across" with some other term to make meaning clearer

Line 203 - Is Figure 2a the correct figure to refer to here?

Line 213 - "cells" is used here to mean two different things. Maybe another word could be used to make the sentence clearer?

Line 235 - "The numbers according to 70 drugs and their target are shown in Table 1" does not make sense

Author Response

This manuscript describes a method for high throughput evaluation of drugs. The method allows evaluation of the effect on cell viability and migration of drug treatments. The authors have developed a micropillar/microwell system that, in combination with calcein staining and image analysis, can evaluate the effect of drugs on cell morphology and hence migration and viability. They screened 70 drugs and identified migration inhibiting drugs. Four of these drugs were validated using conventional migration assays.

Questions of interest:

Why do the cells migrate through the matrigel towards the pillar? Does it have anything to do with gravity?

Why do the chips need to be inverted for drug treatment?

Response) As reviewer’s comments, we think gravity is main reason for cell to migrate the surface of the pillar. Initially, we collect the cells at the curved end of the Matrigel spot. During culture, pillar and well chips were inverted. Over time, cells migrate through the Matrigel and attach to the surface of the pillar by gravity, as shown in Figure 1a. We revised this sentence in 58 lines and Figure 1a.

Line 114 seems to imply that all the drugs were added at 20uM, why would this be the case?

Response) Migration assay is short-term assay culturing cell for 24h. So, we choose 20uM high concentration in all compounds. We also considered that cells embedded Matrigel are generally highly resistant compounds. We add these sentences in 117 lines.

The manuscript is mostly well written and clearly describes the work carried out. The data presentation and figure preparation are excellent. The figures really aid understanding. In order to make the meaning even easier to understand some changes are recommended:

The introduction would benefit from the use of some paragraphs

Response) We revised manuscript.

Lines 44-50 - Due to use of tenses and phrasing used it's difficult to understand if this is previous work in the field, generally known facts or opinion

Response) In Lines 44-50, it is previous work in the the field. So, we unified in the past tense.

Line 56-57 - The meaning of this sentence is unclear

Response) We remove unclear sentence.

Line 68 - Does this mean four migration-inhibiting drugs that were already known? Four classes of migration-inhibiting drugs? Four of the migration-inhibiting drugs with the most easy to observe effects?

Response) We revised confused sentence as “To prove the proposed 3D cell migration assay, conventional migration assays were per-formed using transwell for verification of the four migration-inhibiting drugs which were found in the proposed 3D cell migration assay.

Figure 1 - Replace "Slide Glass" with "Glass microscope slide". I suggest changing "Icing for cell down And Matrigel Gelation" with "Chilling for cell sinking and Matrigel gelation". In the legend it might be helpful to explain what the cells were stained with

Response) We revised Figure 1 and legend.

Methods

Line 88 - centrifuge speed is reported in rpm, this should be reported in g

Response) We revised manuscript.

Line 99 - Please give details of plasma treatment

Response) We revised manuscript.

Line 101 - Please report final concentration of laminin. Where was the laminin from?

Response) We revised manuscript.

Line 115 - please give details of final DMSO concentration in growth media

Response) We revised manuscript.

Line 121 - Icing means "a sweet, creamy spread, as of confectioners' sugar, butter, and flavouring, for covering cakes, cookies" I suspect "chilling" would be a better word to use. "migrate" also does not seem to be a good word to use as it tends to suppose an active process, whereas the cells probably fall through gel to settle at the bottom

Response) We revised manuscript as reviewer’s recommend.

Line 123 - replace incubation with incubator

Response) We revised manuscript as reviewer’s recommend.

Line 126 - please give supplier and concentration of calcein AM used

Response) We revised manuscript.

Line 157-158 - Why was the chip dried for 2 hours after staining? This seems like a long time for a live cell stain to remain on cells

Response) During staining cells, cells are dead but the morphology is remained. The fluorescence is remained during 2 hours in darkness in our previous work. We add reference.

Results and Discussion

Line 199 - Would suggest using "high roundness value" as "high roundness" is confusing. To me it means that highly rounded cells have migrated, not that cells with a high value for roundness have migrated

Response) The roundness of circle is 0. Thus, a high roundness value indicates a migrating spread of cells. We add these sentence in 207 line.

Line 202 - Replace "spread across" with some other term to make meaning clearer

Response) We move confused words.

Line 203 - Is Figure 2a the correct figure to refer to here?

Response) We revised as Figure 3b.

Line 213 - "cells" is used here to mean two different things. Maybe another word could be used to make the sentence clearer?

Response) We revised unclear sentence as “Seventeen cells in the quadrant 1 of Figure 2c had roundness greater than 0.2 and cell area greater than 500 pixels. These cells having high roundness and cell area indicate mi-grated cells.”

Line 235 - "The numbers according to 70 drugs and their target are shown in Table 1" does not make sense

Response) We revised unclear sentence as “The 70 drugs and their target are shown in Table 1”

Reviewer 2 Report

In this article, the authors present a novel micro-array assembly of micropillars and microwells to conduct cell migration experiments. Cells are seeded in Matrigel, on micropillars which are coated with laminin. The micropillars are then placed inverted into the microwells containing media and the drug to be tested. After culturing the cells for a certain amount of time, they are stained and then imaged. A study of the roundness and area of the cells in the image is done, and then plotted on a graph to assess migration and viability.

Below are some comments, corrections or clarification that could improve the article:

Clarifications or corrections:

-          In cell preparation, the resuspension of cells was done in RPMI media supplemented with 10%FBS, but not with 1% penicillin/streptomycin. Why?

-          Why are the microwell/micropillar chips more suitable for mammalian cell lines as claimed in line 95? Explain or reference.

-          Lines 118-119: 950 nL were dispensed in the microwell?

-          Roundness calculation utilized is unclear. What is considered the object area? An example in the figure, or as supplementary information/figure would be helpful.

-          In figure 2a, it might be good to explain in the caption why no cell-stained image is used for no migration inhibition with high toxicity quadrant.

-          The confidence interval is 95%, which correlates to a significance value of p=0.05. Why is the p value taken to be 0.01?

SStatistical analysis should be performed using ANOVA not t-test

The term Roundness is really confusing, perhaps you would want to chenge that to migration or area.

In section 3.1, it is said that having a roundness score closer to 0 means that the shape is close to a circle however in lines 205-206, it is asserted that the cells showing high roundness and high area are in quadrant 1. If high roundness means closer to 0, and high area means a bigger number, then the cells would be in quadrant IV  as per the cartesian norm of numbering quadrants. If a different numbering method is used, please label the quadrants on the figure for at least 1 graph. Or did you mean figure 3b? Please check the figure references throughout the manuscript.

-          In Table 1, what do the columns #1, #2 and #3 represent? Please explain. If these are repeats (n=3) please indicate and add in the statistics section. 

It might be good to add a bar graph or some other way of describing the data that is more clear than the table, in case the table is your preference please use a way to identify the significant samples with a different color or an asterisk or other indicators. 

English language errors:

-          Line 47: the “E” in “Extracellular” should be lowercase

-          Line 80: the sentence should read “… cultivated IN RPMI media…”

-          Line 113: replace duplicates with replicates

-          Line 168: is it JPGE or JPEG image?

-          Line 241: capitalize Vandetanib

-          Table 1: Several drugs are not capitalized in the relevant column. Also, in columns “#1” and “#2” and “#3”, Cell area is misspelled. Drug 63 should be renamed to remove the number. Also, Drug 56 target has a typographic error (smoothen).

Author Response

In this article, the authors present a novel micro-array assembly of micropillars and microwells to conduct cell migration experiments. Cells are seeded in Matrigel, on micropillars which are coated with laminin. The micropillars are then placed inverted into the microwells containing media and the drug to be tested. After culturing the cells for a certain amount of time, they are stained and then imaged. A study of the roundness and area of the cells in the image is done, and then plotted on a graph to assess migration and viability.

Below are some comments, corrections or clarification that could improve the article:

Clarifications or corrections:

-          In cell preparation, the resuspension of cells was done in RPMI media supplemented with 10%FBS, but not with 1% penicillin/streptomycin. Why?

Response) We missed 1 % penicillin-strephtomycin. We revised the manuscript.

-          Why are the microwell/micropillar chips more suitable for mammalian cell lines as claimed in line 95? Explain or reference.

Response) We revised sentence and add reference as “this system is more robust and flexible for use in mammalian cell cultures, enzymatic re-actions, viral infection, and compound screening [9-11].”

-          Lines 118-119: 950 nL were dispensed in the microwell?

Response) We revised sentence and add reference as “the Cell Spotter dispensed 50 nL droplets of the cell/Matrigel mixture on the micropillar and 950 nL of media or drugs in the microwell, respectively.”

-          Roundness calculation utilized is unclear. What is considered the object area? An example in the figure, or as supplementary information/figure would be helpful.

Response) We revised Figure 2a using cell image as example.

-          In figure 2a, it might be good to explain in the caption why no cell-stained image is used for no migration inhibition with high toxicity quadrant.

Response) We revised legend of Figure 2a

-          The confidence interval is 95%, which correlates to a significance value of p=0.05. Why is the p value taken to be 0.01?

Response) As referee’s comment, we change the confidence interval is 99%.

Statistical analysis should be performed using ANOVA not t-test

Response) We re-calculate P-valve using ANOVA and revised statistical analysis, Figure 3c, and Table 1.

The term Roundness is really confusing, perhaps you would want to change that to migration or area.

Response) Roundness is conventional parameter of image analysis. So, we reinforced the roundness description as shown in the modified figure 2a.

In section 3.1, it is said that having a roundness score closer to 0 means that the shape is close to a circle however in lines 205-206, it is asserted that the cells showing high roundness and high area are in quadrant 1. If high roundness means closer to 0, and high area means a bigger number, then the cells would be in quadrant IV as per the cartesian norm of numbering quadrants. If a different numbering method is used, please label the quadrants on the figure for at least 1 graph. Or did you mean figure 3b? Please check the figure references throughout the manuscript.

Response) We revised confused sentences and Figure 2a.

-          In Table 1, what do the columns #1, #2 and #3 represent? Please explain. If these are repeats (n=3) please indicate and add in the statistics section. 

Response) #1, #2, and #3 mean chip number. We revised indication in statistic section.

It might be good to add a bar graph or some other way of describing the data that is more clear than the table, in case the table is your preference please use a way to identify the significant samples with a different color or an asterisk or other indicators. 

 Response) Based on the Table 1, we make Fig.3b. We revised table 1 by marking significant samples with color.

English language errors:

-          Line 47: the “E” in “Extracellular” should be lowercase

-          Line 80: the sentence should read “… cultivated IN RPMI media…”

-          Line 113: replace duplicates with replicates

-          Line 168: is it JPGE or JPEG image?

-          Line 241: capitalize Vandetanib

-          Table 1: Several drugs are not capitalized in the relevant column. Also, in columns “#1” and “#2” and “#3”, Cell area is misspelled. Drug 63 should be renamed to remove the number. Also, Drug 56 target has a typographic error (smoothen).

 Response) We revised manuscript as referee’s comments.